# Microbial Diversity Dynamics in a Methanogenic-Sulfidogenic UASB Reactor

**DOI:** 10.3390/ijerph18031305

**Published:** 2021-02-01

**Authors:** E. Fernández-Palacios, Xudong Zhou, Mabel Mora, David Gabriel

**Affiliations:** GENOCOV Research Group, Department of Chemical, Biological and Environmental Engineering, Escola d’Enginyeria, Universitat Autònoma de Barcelona, 08193 Bellaterra, Spain; eva9520@gmail.com (E.F.-P.); Xudong.Zhou@uab.cat (X.Z.); mabel.mora@uvic.cat (M.M.)

**Keywords:** bioscrubber, crude glycerol, UASB, sulfate-reducing bacteria, microbial diversity shifts, sequencing

## Abstract

In this study, the long-term performance and microbial dynamics of an Upflow Anaerobic Sludge Blanket (UASB) reactor targeting sulfate reduction in a SOx emissions treatment system were assessed using crude glycerol as organic carbon source and electron donor under constant S and C loading rates. The reactor was inoculated with granular sludge obtained from a pulp and paper industry and fed at a constant inlet sulfate concentration of 250 mg S-SO_4_^2−^L^−1^ and a constant C/S ratio of 1.5 ± 0.3 g Cg^−1^ S for over 500 days. Apart from the regular analysis of chemical species, Illumina analyses of the 16S rRNA gene were used to study the dynamics of the bacterial community along with the whole operation. The reactor was sampled along the operation to monitor its diversity and the changes in targeted species to gain insight into the performance of the sulfidogenic UASB. Moreover, studies on the stratification of the sludge bed were performed by sampling at different reactor heights. Shifts in the UASB performance correlated well with the main shifts in microbial communities of interest. A progressive loss of the methanogenic capacity towards a fully sulfidogenic UASB was explained by a progressive wash-out of methanogenic *Archaea*, which were outcompeted by sulfate-reducing bacteria. *Desulfovibrio* was found as the main sulfate-reducing genus in the reactor along time. A progressive reduction in the sulfidogenic capacity of the UASB was found in the long run due to the accumulation of a slime-like substance in the UASB.

## 1. Introduction

Sulfur is one of the most abundant elements on Earth, mainly present in sediments and rocks in the form of sulfate minerals (gypsum, CaSO_4_), sulfide minerals (pyrite, Fe_2_S), and sulfur deposits (S^0^), which have been formed in different geological periods. Sulfur is a pale yellow, odorless, and insoluble chemical element with symbol S and atomic number 16. However, hydrogen sulfide (H_2_S), the last sulfur compound of the reduction path, is a volatile, toxic, corrosive, malodorous compound that causes an impact on the chemistry of the environment, also due to its reactivity with metals. Therefore, the formation of this volatile, dissolved or gaseous, end-product is not desirable in the sulfur cycle, contrary to what happens in both carbon and nitrogen cycles, for example.

Since the Industrial Revolution, the increasing amount of human activities, especially through the combustion of fossil fuels and the processing of metals, have contributed to the amount of sulfur entering into the atmosphere. Combustion of sulfur-containing fuels, such as coal, natural gas, peat, wood, and oil, results in SO_2_ formation mainly generated in the energetic and industrial sectors [1]. Emissions from these activities, along with nitrogen emissions, react with other chemicals in the atmosphere to produce tiny particles of sulfate salts, which fall as acid rain and can cause health impacts, acid deposition in the environment, and visibility depletion [2]. Air pollution is not only a matter of climate change but also its negative impact on the public and individual health should be highlighted. Among others, nitrogen oxide, sulfur dioxide, Volatile Organic Compounds (VOCs), and dioxins are all considered air pollutants that are harmful to humans. Some of the most common diseases occurring from the aforementioned substances include respiratory problems, asthma, bronchiolitis, lung cancer, cardiovascular events, central nervous system dysfunctions, and cutaneous diseases [3]. Even if many studies have been carried out on this topic, many questions about the impact of SO_2_ and other sulfur oxides on human health remain unclear [4].

Flue gases are usually treated through physical-chemical processes. One of the most implemented treatment to remove SO_2_ from flue gases is wet flue gas desulfurization (WFGD). However, these processes are expensive (due to the use of alkaline absorbents) and generate additional effluents requiring further processing and energy inputs [5,6]. The development of environmentally friendly alternatives to valorize not only SO_2_ from WFGD but also S-rich liquid effluents are clearly needed. The two-stage bioscrubber concept described in [7] is a promising integrated process to recover elemental sulfur as a value-added product. The process consists of a first scrubbing stage of SO_2_ using slightly alkaline absorbents, followed by a two-stage biological process: (i) Reduction of sulfite and/or sulfate to sulfide and (ii) partial oxidation of sulfide to elemental sulfur. The implementation of an S valorization-based process pursues treating wastewater much more efficiently in terms of energy consumption. Nowadays, the transition towards a circular economy is becoming a priority in the EU, and it can be an opportunity to promote the conversion of wastes to value-added products; and, therefore, to enhance the efficiency of resource utilization. In the actual bio-economy era, the establishment of the concept of the circular economy would expand and diversify the market of bio-based products as bio-based chemicals, biopolymers, fuels, and bio-energy [8]. Elemental sulfur is currently obtained from the petrochemical industry, thus, its recovery has strong potential from a sustainable and environmental point of view.

Anaerobic treatment of wastewaters containing significant amounts of sulfate presents a challenge due to competition between sulfidogenesis and methanogenesis. Sulfate reduction during the anaerobic treatment of wastewaters is generally an unwanted process because of the reduction in methane yield and problems of corrosion and toxicity caused by H_2_S. The production of hydrogen sulfide by sulfate-reducing bacteria (SRB) can be toxic to the various trophic groups of bacteria involved in the process [9]. Therefore, research efforts have been commonly focused on its negative role in anaerobic wastewater treatment, studying H_2_S toxicity and competition between methanogens and sulfate reducers to suppress sulfidogenesis [10]. To what extent sulfate reduction will predominate over methanogenesis depends on many factors, including the organic substrate to sulfate ratio (COD/SO_4_^2−^) of the wastewater and the type of organic substrate, among others.

Over the last decades, research on microbial communities’ evolution in engineered biosystems has gained interest since it is of high importance to get them controlled and continuously enhanced. For a long time, biological reactors have been considered as a “black box” where microbiological phenomena taking place were not elucidated [9]. This indifference was mainly due to the lack of analytical techniques able to identify those microorganisms playing a role in the biodegradation processes. During the 1990s, a new set of molecular techniques transformed microbial ecology research [11], overcoming the scarcity of information provided by traditional methodologies. As a result, the most broadly used techniques nowadays are fluorescent in situ hybridization (FISH), cloning or sequencing of 16S rRNA, and denaturing gradient gel electrophoresis (DGGE). There is a clear trend among the literature to combine molecular techniques as well as microbiological methods in engineered ecosystems [12,13,14]. This is important to reduce potential biases and limitations of the different techniques, and hence to obtain a more realistic picture of microbial community structures and their functionality.

A previous study [15] highlighted the potential of crude glycerol utilization as a possible carbon source to reduce sulfate in an anaerobic bioreactor (UASB reactor) treating sulfate laden wastes. Nevertheless, almost no information about the links between the microbial community structure and the bioreactor performance is currently available. Therefore, the purpose of this study was to: (i) Evaluate the microbial stratification in the sulfidogenic UASB reactor under a low up-flow velocity regime (≈0.25 m h^−1^), (ii) examine the temporal dynamics of the microbial populations of the UASB fed at a constant TOC/S ratio, and (iii) correlate physical-chemical parameters with microbiological changes to explain better the dynamics of the process. A better understanding of the biological processes taking place in the UASB reactor operated is anticipated to be gained in this study.

## 2. Materials and Methods

### 2.1. Reactor Set-Up and Operation

A 2.5 L lab-scale UASB reactor was used for the long-term operation period (550 days) in this study. The detailed diagram of the reactor as well as the experimental setup details and the different conditions used, are described in [15]. In short, the composition of the mineral medium was (g∙L^−1^): K_2_HPO_4_ (3), NH_4_Cl (0.2), and Na_2_SO_4_ (1.15) dissolved in tap water to add macro- and micronutrients and adjusted to pH = 8.8–9.0 with NaOH (2 M). The buffering capacity of the mineral medium allowed maintaining the outlet pH around 7 along with the whole operation of the reactor (Appendix A). No chemicals were applied to remove dissolved oxygen in the synthetic medium. The reactor was operated at a constant sulfate inlet concentration of 253 ± 21 mg S-SO_4_^2−^∙L^−1^, therefore, the sulfate loading rate (SLR) was 5.1 ± 0.7 kg S-SO_4_^2−^ m^−3^∙d^−1^. A constant organic loading rate (OLR) of 7.6 ± 1.6 kg C m^−3^∙d^−1^ was set in order to obtain a steady-state operation and a constant TOC/S ratio of 1.5 ± 0.3 g C g^−1^∙S, thus minimizing the effect of changing conditions. Crude glycerol (Appendix A) produced in the biodiesel industry was used throughout the whole operation as a carbon source. Inlet and outlet flows were sampled every 2–3 days to perform chemical analyses. The inoculum, 1 L of granular sludge, was obtained from an anaerobic digester treating wastewater in a pulp and paper industry. The hydraulic residence time (HRT), calculated as that corresponding to the reaction volume only (sludge blanket), ranged between 1 h and 1.8 h. Moreover, inoculum and biomass samples collected along the reactor operation were investigated to analyze shifts in the different microbial populations through Illumina sequencing analysis.

### 2.2. Analytical Methods

Ion chromatography with conductivity detection in a Dionex ICS-2000 equipment with an Ultimate 3000 Autosampler Column Compartment and an IonPac AS18 column (Dionex, Sunnyvale, CA, USA) were used for sulfate (SO_4_^2−^) and thiosulfate (S_2_O_3_^2−^) concentrations monitoring. Total inorganic (TIC) and total organic carbon (TOC) were analyzed in a multi N/C 2100S (Analytik Jena, Jena, Germany). A selective sulfide electrode (VWR, Avantor, Radnor, PA, USA) connected to a benchtop meter (Symphony, VWR, Avantor, Radnor, PA, USA) was used for the off-line measurement of the total dissolved sulfide (TDS) concentration. Samples pre-treatment and dilution were performed as described in [13]. Volatile fatty acids (VFAs) were determined in a Dionex 3000 ultimate high-performance liquid chromatography (Dionex, Sunnyvale, CA, USA) using a UV detector at 210 nm. The chromatographic separation was performed in an ICE-COREGEL 87H3 column (7.8 × 300 mm, Transgenomic, Omaha, NE, USA), heated at 40 °C, employing 0.006 mM of H_2_SO_4_ as a mobile phase at a flow rate of 0.5 mL min^−1^. Samples were filtered at 0.22 µm (Millipore, Burlington, MA, USA). 

CH_4_ and CO_2_ contained in the biogas were analyzed by gas chromatography (7820 A, Agilent Technologies, Wood Dale, IL, USA). The volume of the gas produced in the UASB reactor was calculated following the Gas Bag Method (GBM) as described in [16]. In short, the average methane flowrate was calculated based on the volume of gas collected along variable time periods in which biogas was accumulated in the sampling bag located on top of the UASB and the methane concentration in the gas bag. H_2_S production was analyzed by gas chromatography (Hewlett-Packard HP 5890 A, Agilent Technologies, Wood Dale, IL, USA) using a thermal conductivity detector and a Porapak Q column with helium as a carrier gas.

### 2.3. Illumina Sequencing Analysis

Identification of the microbial population was performed using the Illumina platform, on different samples, along with the operation of the reactor (Table 1), at different reactor’s heights (Appendix A) and including the inoculum. Different UASB heights were sampled at different operation times because the sludge bed was not static due to flotation. Therefore, obtaining sludge from the same sampling points was not always possible. Moreover, this way, the stratification of the sludge could be studied and related to the results of the physical-chemical profiles obtained also at different heights. Genomic DNA of all samples was extracted by applying the protocol of PowerSoil™ DNA isolation kit (MoBio Laboratories, Carlsbad, CA, USA) following the supplier’s instructions. The quantity and quality of the extracted DNA were assessed by using a NanoDrop 1000 Spectrophotometer (Thermo Fisher Scientific, Waltham, MA, USA), and then DNA samples were preserved at −20 °C for further analysis.

For samples collected along the operation (Table 1), sequencing analyses were performed by “Genomic and Bioinformatics service” at the UAB University. For library preparation, a fragment of the bacterial 16S V3-V4 ribosomal RNA gene of around 400 bp was amplified using the primers 341F (5′ CCTA CGG GNG GCW GCA G 3′) and 805R (5′ GAC TAC HVG GGT ATC TAA TCC 3′) according to [17] as the most promising bacterial primer pair. The database used for the classification of organisms was based on the Greengenes database (http://greengenes.lbl.gov/). Apart from the samples in Table 1, a sample of the inoculum was collected to also perform Illumina sequencing analysis. This time, sequencing was performed using a MiSeq System by an external service (Scsie UV, Valencia, Spain). The high throughput sequencing was carried out by amplifying the V3-V4 hypervariable region of 16S RNA gene of the extracted DNA with the universal primers 341F (5′-CCT ACG GGN GGC WGC AG-3′) and 806R (5′-GGA CTA CHV GGG TAT CTA AT-3′).

## 3. Results

### 3.1. Long-Term Performance of the UASB at Constant Loading Rate

The long-term performance of the UASB was evaluated at a constant sulfate loading rate (SLR) during 550 days of continuous operation in terms of sulfate removal efficiency (S-RE), TOC removal efficiency (TOC-RE), and sulfate and TOC elimination capacities (S-EC and TOC-EC, respectively). Table 2 shows the results obtained from the long-term UASB operation as averages and standard deviations of all data acquired. The operation was divided into three different periods according to the stability of the performance. Period I (day 0–250) corresponds to the start-up of the reactor together with the period in which the reactor was performing positively, obtaining satisfactory removal efficiencies; Periods II and III correspond to a progressive decline in the removal efficiencies leading to an unstable and low-performing operation.

Results of the monitoring of sulfur species are shown in Figure 1, while TOC and the concentration of each VFA monitored together with the average flowrate of methane can be observed in Figure 2.

The TOC/S ratio was almost constant during the whole operation with the intention of keeping the more possible stable conditions in terms of carbon and sulfate inlet loading rates, thus allowing the reactor and microbial populations to adapt and develop without changing scenarios. As shown in Table 2, a clear difference can be seen between each operating period in terms of efficiencies, even if inlet conditions were constant throughout the whole operation. Period I represents the start-up of the reactor together with a promising performance; Period II is a transition period between the good performance and the underperformance of the operation confirmed in period III. As can be observed in Figure 1, sulfate reduction started almost immediately after the start-up of the reactor. Outlet sulfate concentration decreased gradually until reaching an almost steady-state along the first 250 days of operation. From day 0 until day 250 (period I), the average S-RE was 80 ± 17%. From day 251 until day 400 (period II), the efficiency decreased to 52 ± 15%, while in period III, it was 33 ± 8% on average, indicating a significant sulfate efficiency loss. TOC was almost completely consumed during the first 135 days of operation, obtaining concentrations below 43.4 mg C∙L^−1^ in the effluent (Figure 2A). However, a reduction was observed in the TOC-RE that decreased from 70% in period I to 16% in period II. The S-EC during period I was 4.4 kg S m^−3^∙d^−1^. However, the S-EC decreased down to 2.5 kg S m^−3^∙d^−1^ and 1.6 kg S m^−3^∙d^−1^ in periods II and III, respectively. This significant decrease of the TOC-RE was coupled to a progressive VFA accumulation, especially acetate (Figure 2B). At the same time, the increase in acetate concentration coincided with a significant decrease in the CH_4_ production (Appendix A) after 135 days of the start-up of the operation. On day 150, the flow of methane had almost ceased completely and sulfide concentration by that day was 200 mg S∙L^−1^. The average biogas composition during the first 4 months of operation, when biogas was produced in a relatively stable flow, was (%v/v) 84.6 ± 8.2%, 10.3 ± 2.1%, and 1.8 ± 0.6% for CH_4_, CO_2,_ and H_2_S, respectively.

### 3.2. Microbial Diversity Changes and Stratification in the Sulfidogenic UASB 

#### 3.2.1. Long-Term Performance and Microbial Community Analysis

Samples were collected throughout the whole operation, coinciding with different situations from which Illumina information could be useful to infer conclusions in relation with the UASB performance. Samples from different reactor heights (Appendix A) were collected. However, only samples from UASB 1 were considered for days 149, 173, 230, and 294 since this was the most stable part of the bed of the UASB. In the case of samples from days 85 and 538, samples from UASB 6 were presented. On day 85, when the first sample was collected, there was a significant amount of gas produced (80 mL h^−1^), and increasing S-RE was achieved (82%). Figure 3 shows the relative abundances (%) at the genus level of all the different samples along the operation of the reactor. 

As can be observed, on day 85, *Desulfovibrio* was the most abundant genus detected, with a relative abundance of 17.8%. Such percentage was remarkable considering that the inoculum was obtained from an anaerobic digester where only a 0.02% of relative abundance of genus *Desulfovibrio* was detected. In less than 100 days, this genus was able to increase its relative abundance noticeably. *Syntrophobacter* and *Propionispora* were the following most abundant genus detected in this sample with a relative abundance of 8.6% and 5.1%, respectively. On day 149, both TOC-RE and S-RE were the highest of the different sampling events, 91% and 72%, respectively, while methane production had decreased significantly by that time (Figure 2B) compared to its production at the beginning of the UASB operation. On the other hand, acetate had started to accumulate in the system, reaching a concentration of 292 mg∙L^−1^ on day 149. On that day, *Desulfovibrio* was the most abundant genus detected, with a relative abundance of 36.2%. *Propionispora*, *Syntrophobacter,* and *Aminiphilus* were the next ones in order of relative abundance with a 5.9, 5.4, and 5.2%, respectively. On day 173, acetate concentration was 380 mg∙L^−1^ and the TOC-RE had decreased to 44%. According to the sequencing results on that day, *Desulfovibrio* was the most abundant genus detected with a 42.8%, the highest percentage among all the samples considered.

From day 230 onwards, acetate accumulation was considerable, reaching its highest concentration on day 253 (527 mg∙L^−1^). From day 230 onwards, no methane was produced. The relative abundance of *Desulfovibrio* by that day was 40.7%. *Propionispora* was the next genus with higher abundance, 13.9%. By day 294, the TOC-RE had decreased significantly to 17%, whereas the S-RE was 50%. The acetate concentration measured that day was 406 mg∙L^−1^. *Desulfovibrio* decreased its relative abundance to 30% and *Dysgonomonas* became the next genus in order of abundance with a 13.6%. On day 538, the UASB was clearly underperforming at an S-RE of 29%. According to the results gathered by Illumina, *Propionispora* was the most abundant genus with a relative abundance of 15.2%, followed by *Dysgonomonas* (13.2%) and *Desulfobulbus* (11.6%). *Desulfovibrio* decreased its relative abundance until 10.8%. 

Having a look at the *Archaea* domain, only *Methanosaeta* genus was detected with a relative abundance higher than 1%. Therefore, this genus is the only one shown in Figure 3. *Methanosaeta* was the most abundant group in the inoculum, but its relative abundance decreased along time, being almost undetectable in samples from day 230 onwards, indicating that these populations were washed-out from the system. To get a better comparison between the two groups of interest in this work, Table 3 shows the results of the relative abundances (%) of all the methanogens detected and considered as a group, together with genus *Desulfovibrio*. The latter genus was selected as it was the main one of all the sulfate-reducers detected during the whole operation. As can be observed from this table, there was a clear decrease in the relative abundance of methanogens and a remarkable increase in the relative abundance of sulfate reducers (*Desulfovibrio*). In the long-term, a decrease in the relative abundance of *Desulfovibrio* was also detected.

Methanogens are a diverse group of microorganisms, and even if substrates that they utilize are very limited, Figure 4 presents the major classes detected to get a better insight among this group. This information was helpful for obtaining an explanation of parameters observed in the long-term operation, such as acetate accumulation. *Methanosaeta* was the most predominant genus in the inoculum, with a relative abundance of 7.5% and was still present until day 173 but not in the sample from day 230. *Methanobacteria* class was the next one in order of abundance detected in the inoculum sample with a 2%. Hydrogenotrophic methanogens, such as the order *Methanomicrobiales*, was only detected in the inoculum sample and with a relative abundance of 0.7%. Figure 4 provides a visual overview of how methanogens were only predominant in the inoculum and how *Desulfovibrio* could quickly colonize the UASB reactor under the operating conditions.

#### 3.2.2. Chemical and Microbial Stratification in the UASB Reactor under Non-Methanogenic Conditions

With the purpose of gaining more knowledge about microbial community changes along the operation with constant sulfate and organic loading rates, samples at different reactor heights were analyzed throughout the performance (Table 1). From all the samples analyzed, only day 173 and day 230 were presented, as they were considered the more representative ones in terms of profiles along the different heights of the UASB reactor. Figure 5 shows the most abundant genus detected on day 173 (Figure 5A) and on day 230 (Figure 5B) at different reactor heights. 

Figure 5A reveals that the most abundant genus on day 173 was *Desulfovibrio* and that there was a considerable decrease from UASB 1 and UASB 2 to UASB 3. The relative abundances in samples UASB 1 and UASB 2 were 42.8% and 49.8%, respectively, whereas in UASB 3, it was only 19.5%. Another difference that can be observed in Figure 5A was the increase in the Operational Taxonomic Units (OTUs) detected and assigned to genus *Syntrophobacter*. The relative abundance of this genus was 6.4% in UASB 1 and 9.9% in UASB 3. *Sphingobacterium* genus also increased its relative abundance from 0.8% in UASB 1 to 5% in UASB 3.

Figure 5B shows the microbial diversity of the most abundant genus detected in samples collected on day 230 of operation (UASB 1, UASB 4, and UASB 6). No significant differences among the relative abundances of genus *Desulfovibrio* can be seen at different heights if compared to the situation already described on day 173. The relative abundances of this genus for UASB 1, UASB 2, and UASB 3 were 40.7%, 36.2%, and 36.1%, respectively. *Propionispora* decreased its relative abundance from the bottom to the top of the reactor: 13.9% in UASB 1; 1.8% in UASB 4; and 5.4% in UASB 6. On the other hand, *Aminiphilus* presented the opposite behavior: 1.3% in UASB 1; 3.7% in UASB 4; and 9% in UASB 6. This same trend was also observed for genus *Desulfobulbus*. Its relative abundances for the different heights were: 0.1% in UASB 1; 0.6% in UASB 4 and 7.5% in UASB 6. Curiously, *Klebsiella* presented a higher abundance in UASB 4 (10%) compared to UASB 1 and UASB 6, 0.3% and 1.1%, respectively.

Figure 6A shows the concentration for the main sulfur species involved (sulfate and sulfide) measured at different reactor heights on day 173. The already mentioned decrease in the relative abundance of *Desulfovibrio* species along the different heights can be supported by the different sulfate reduction rates (SRR) calculated for these heights. As can be observed in Figure 6A, almost all the sulfide was produced in the lowest part of the reactor (until UASB 2). From UASB 3 to the outlet, there was no appreciable sulfide production, which can be related to the fact that in UASB 3 the relative abundance of the major sulfate reducer detected (*Desulfovibrio*) decreased significantly as well. From the inlet port to UASB 1 the SRR was 441 mg S L^−1^∙h^−1^, whereas from UASB 1 to UASB 2 the SRR was 148 mg S L^−1^∙h^−1^ and from UASB 2 to UASB 3 the SRR decreased down to 34 mg S L^−1^∙h^−1^.

Figure 6B shows the concentration of the different sulfur species measured along the different heights on day 230. What stands out from this figure was that, to reach the same final concentration of sulfide in the effluent, 192 mg S L^−1^ and 195 mg S L^−1^ on day 173 and 230, respectively, the profiles were quite different. Whereas on day 173, almost the highest sulfide concentration was already reached at UASB 2, a different progression was observed on day 230, where this concentration was not reached until the upper part of the reactor, meaning that sulfate reduction was also occurring in UASB 5–6 (upper part of the reactor).

On day 230, the SRR was 338 mg S L^−1^∙h^−1^ from the inlet port to UASB 1. The SSR was reduced to 94 mg S L^−1^∙h^−1^ from UASB 1 to UASB 2, while the SRR was only 9 mg S L^−1^∙h^−1^ from UASB 2 to UASB 3. Then a huge sulfide production was detected again from UASB 3 to UASB 4 (117 mg S L^−1^∙h^−1^). Figure 6C shows the crude glycerol removal along the reactor´s heights on day 173 and on day 230 to compare the behavior between both operation dates. On day 173 glycerol was not detected from UASB 2 upwards. On the contrary, on day 230, glycerol was detected (49 mg L^−1^) even in the effluent of the reactor, meaning that a gradual drop in the fermenting capacity was also taking place in the reactor.

## 4. Discussion

### 4.1. Long-Term Performance and Microbial Community Analyses

Flue gases are usually treated through physical-chemical processes such as wet flue gas desulfurization (WFGD), one of the most implemented treatment to remove SO_2_ from flue gases. However, these processes are expensive (due to the use of alkaline absorbents) and generate additional effluents requiring further processing and energy inputs [5,6]. The UASB system studied herein presents a crucial stage to valorize not only SO_2_ from WFGD but also S-rich liquid effluents in general as part of the two-stage bioscrubber concept described in [7]. The environmentally-friendly integrated system presents an alternative to recover elemental sulfur as a value-added product. Elemental sulfur is currently obtained from the petrochemical industry, thus, its recovery has a strong potential from a sustainable and environmental point of view. Possible applications for the recovered elemental sulfur could be the pigments industry, which utilizes a wide diversity of chemical compounds in innumerable formulations, or the sulfur fertilizers market.

Taken together, results presented in this study provide the evolution of the operation both in terms of physical-chemical parameters and changes in the microbial communities. The evolution of the operation according to sulfate and organic carbon removal efficiencies and the different parameters measured along the UASB operation are provided in results Section 3.1. 

The UASB performance showed a progressive decrease in the methane production over the first 200 days. This result is consistent with Illumina sequencing results. Table 3 confirms the wash-out of methanogens, not detected from day 173 onwards. The most striking result emerging from Table 3 is the increase in the relative abundance of *Desulfovibrio*, which raises from almost 0% in the inoculum up to its maximum, 42.8% on day 173. Equivalently, Figure 3 shows the disappearance of the genus *Methanosaeta* (the major methanogen detected) that was only present in the inoculum. In our case, the interest was mainly focused on methanogens and sulfate reducers and how their relative abundances shifted completely during the long-term performance. For that reason, Figure 4 presents only the relative abundance (%) of the detected methanogens together with the genus *Desulfovibrio*, as this one became the major sulfate reducer along the operation. All known methanogens belonged to the *Euryarchaeota* phylum; within this phylum, the classes *Methanobacteria*, *Methanococci*, *Methanofastidiosa*, *Methanomassillicocci*, *Methanomicrobia*, and *Methanopyri* are methanogenic. However, recently there has been genomic evidence that within *Bathyarchaeota* and the novel phylum *Verstraetearchaeota*, methane production also occurs [18]. Results obtained herein suggested that sulfate-reducers could outcompete methanogens during the performance. Therefore, methanogens were washed-out from our system, whereas sulfate-reducers (*Desulfovibrio* genus) became the major group detected. This indicated that, in the long-term, both populations were not able to coexist in this UASB under the conditions tested. In general, sulfate reducers always predominate in carbon source utilization and electron flow transmission and suppress the activity of methanogens [19,20]. The population structure determined by Illumina sequencing could be linked to the functional changes observed along the operation in the reactor, in this case, the rate of methane production. However, the in situ metabolic functions of the microorganisms in the UASB were not characterized. The competition of sulfate-reducing and methanogenic populations in anaerobic reactors and in the presence of non-limiting sulfate concentrations have been studied previously [21,22,23,24]. Most of these studies were performed with granular biomass or attached-growth reactors, thus factors such as microbial adhesion and colonization or mass transfer limitations become a crucial factor affecting the competition between these populations [25]. 

The information available on sulfide toxicity and the mechanism of toxicity is frequently ambiguous. It has been reported that the undissociated sulfide molecule is absorbed into the cell and destroys the bacterial proteins, thereby making the cell inactive [26,27]. If this is the case, bacteria should not be able to restart its activity once sulfide is removed. By contrast, it was reported that the sulfide inhibition is reversible, and the normal cell growth and sulfate reduction rates are attained as soon as sulfide is removed from inoculated bioreactors [28,29,30]. In addition to the uncertainty and contradictory information among the literature with respect to inhibitory mechanisms of sulfide, contradictory reports exist with respect to inhibitory effects of various forms of sulfide [31]. Therefore, it is not easy to compare the inhibitory/toxic values reported in different studies, as the inhibition has been assessed based on growth, substrate degradation, sulfate reduction, or cellular yield. The present study did not consider so many parameters but demonstrated that at a TOC/S ratio between 1.4–1.7 g C g^−1^ S, SRB had a competitive advantage over methanogens and that, after 200 days of operation, methanogens were washed-out from the system.

The significant decrease in CH_4_ production after 135 days of the start-up of the operation, which has been previously discussed, coincided with an increase in the acetate concentration. This acetate accumulation and low methane production were also observed by [18]. As can be observed in Figure 3 and Figure 4, *Methanosaeta* was the most abundant genus among the phylum *Archaea*. *Methanosaeta,* together with the genus *Methanosarcina,* have been described as acetoclastic methanogens that use acetic acid as a carbon source to produce methane directly [32]. The almost complete disappearance of these groups from day 85 onwards (when the relative abundance of *Methanosatea* was 2.7%) was related to the decrease in the methane production rate. At the same time, this fact can explain the increase in acetate concentrations. After the washout of these populations, the system began to accumulate this metabolite as acetoclastic methanogens were the ones in charge of using acetate to produce methane. Furthermore, hydrogenotrophic methanogens, such as the order *Methanomicrobiales*, can produce methane indirectly from acetate, which is converted into H_2_ and CO_2_ and further to methane [33]. However, this order was only detected in the inoculum sample with a relative abundance of 0.7%, which also supports the conclusion that there were no populations able of using this acetate, that ended up accumulating in the system.

Glycerol was also detected in the effluent from day 225 (32 mg L^−1^) onwards, reaching concentrations of 362 mg L^−1^ on day 550. Figure 6C confirmed that, by day 173, glycerol was consumed before reaching UASB 2, however, on day 230, glycerol was detected even at the outlet of the reactor. These results suggest that crude glycerol could not even be completely fermented or converted to other easily biodegradable compounds. From day 230 onwards, the relative abundance of the OTUs assigned to the genus *Desulfovibrio* decreased significantly until the end of the operation. That may be related to the decay in the performance of the reactor and the almost complete loss of sulfate removal capacity. Another possible explanation for the decay in the overall performance could be due to the fact that a huge biofilm or slime-like substance was covering the walls of the reactor, as can be observed in Appendix A. Dissolved oxygen could result in sulfide oxidation in the outlet of the reactor, causing sulfur deposits attached to the walls next to the outlet tube, as can be observed in Appendix A. However, the slime-like substance was observed along the whole sludge blanket. Preliminary results of further characterization of the slime-like substance indicate that its nature is mostly organic matter (results not shown). This slime was probably conferring properties such as viscosity to the sludge and, consequently, problems related to mass transfer limitations, affecting as well, the sulfate-reducing activity of the granules and leading to a decrease in the S-RE. The gelatinous and sticky nature of the slime attached to the surface of the granules and to the reactor walls may be coming from the impurities contained in crude glycerol. A loading rate corresponding to a concentration lower than 250 mg S-SO_4_^2−^ L^−1^ has never been tested, thus no conclusions about the operation at this load can be made. However, from experience obtained through several operations, the failure could also occur under these conditions as the overgrowth of this slime is not directly related with the sulfate load but we suspect that the slime is accumulated due to the composition of the crude glycerol. The C/S ratio of 1.5 ± 0.3 g C g^−1^ S was selected according to the best results obtained from the operation described in [15]. As observed by the authors, once the slime was fully present in the sludge, no changes (neither C/S ratio nor up-flow velocity) could make the system work properly again. The concentration of Volatile Suspended Solids (VSS) measured on UASB 3 was between 20 and 30 g L^−1^, whereas it dropped from 58 g L^−1^ (78 days) to 31 g L^−1^ (390 days) on UASB 1 as shown in Appendix A. Together with the accumulation of slime observed towards the end of the operation (after day 330), it was also noticed that part of the sludge floated from the bottom to the upper part of the reactor (UASB6). Overall, these results suggested that not only sulfate reducers were being affected, but the whole system was not performing properly. Further studies are currently being conducted, such as analysis through a mix of physical-chemical and microbial analyses of the slime-like substance and testing the operation of the reactor under the same conditions but using an internal recirculation flow from top to bottom of the reactor to increase the liquid velocity. Therefore, trying to minimize the accumulation of slime thus as to test if higher up-flow velocities could avoid this failure in the long-term and further determine to which extend the slime could be affecting the operation.

### 4.2. Stratification of Microbial Communities and Chemicals

Samples for 16S sequencing were not collected in triplicates, thus no statistical analysis could be performed. Nevertheless, as reported by [34], bias and variability inherent to the PCR amplification and sequencing steps are significant enough to hide differences between bacterial communities from replicate samples. PCR amplification and sequencing errors have been considered inconvenient for 16S rRNA gene amplicon [35,36]. Nevertheless, this approach offers a broad overview for a large microbial community characterization and allows detecting rare species in complex communities. As has already been mentioned, a combination of different molecular biological techniques is the best way to obtain an accurate picture of what is happening during the operation of a bioreactor. Therefore, many authors have reported the use of conventional microbiological methods in combination with kinetic modeling, 16S rRNA gene analyses, FISH, DGGE, or other techniques to get insight into the microbial community of different systems [37,38,39,40]. 16S sequencing is neither a quantitative technique [11] thus, no clear relationship can be made between the relative abundances of the microorganisms involved, the metabolic activities, and sulfide production in this case. However, the comparison between the profile measurements provided us with information about how the reactor was performing.

When having a look at the different samples along the reactor heights in Figure 5A, the most abundant genus found in all samples were the same, as discussed above. However, an increase in the OTUs detected and assigned to the genus *Syntrophobacter* can be observed if UASB 1 is compared to UASB 3. All members of this genus anaerobically degrade propionate to acetate in the presence of methanogens. *Syntrophobacter* is often found in sludge from anaerobic waste treatment facilities and is useful for further degrading organic compounds from propionate and lactate to acetate. If that were the main reaction happening, this fact would also support the accumulation of acetate observed in Figure 2B.

Considering the total amount of sulfide produced, on day 173, 91% had been already produced by UASB 2, whereas on day 230, only a 68% had been produced at the same reactor height. Those results suggested that the reactor was losing sulfate-reducing capacity in the first part of the sludge bed. Even thus, the concentration of sulfide measured in the outlet of the reactor was 226 mg L^−1^ and 195 mg L^−1^ on day 173 and 230, respectively. That would mean that there was not much difference (31 mg L^−1^) on the total amount of sulfide produced. Still, the sulfate reducing activity was not equally being developed in terms of reactor heights.

## 5. Conclusions

Long-term performance of a sulfidogenic UASB reactor under a constant loading rate can be achieved and lead to highly dynamic conditions. Despite such constant feeding and environmental conditions, highly dynamic changes in terms of microbial shifts were observed. Microbial communities specialized in more specific functions, and SRB populations were selected according to operating conditions. The non-acetate degrader *Desulfovibrio* was found to be the most abundant SRB genus detected. The decrease in TOC removal efficiency was linked to the increase in acetate concentration that was related to the washout of methanogens. Physical and chemical parameters and Illumina data correlated well to explain the methanogenesis dynamics. However, microbial diversity dynamics in the long-term could not explain the decrease in sulfate and TOC removal efficiencies. 

Profiles measured along the operation along different heights of the reactor were also helpful in understanding the loss in the fermentative capacity of the system linked to the reduced performance. The performance was hindered by the progressive growth of a slime-like substance that was found to be a crucial factor affecting the structure of the bed and, possibly, leading to problems related to mass transfer limitations. The structure of the bed was affected by the sludge flotation that leads to a VSS drop on the fermentative section of the reactor dropped, an increase of VSS concentration on top of the bed, and a progressive loss of VSS in the reactor after 330 days. Further, analyses of the slime-like substance are warranted to characterize the substance and obtain deeper conclusions about its effect on reactor performance.

It is probably not essential to know the phylogenetic position at a species level of an individual microorganism for the design and operation of a reactor for wastewater treatment. However, a general overview and the follow up of certain populations of interest along the operation help us to relate key factors such as the decrease of methane and the accumulation of acetate with changes in these populations, specifically the washout of methanogens. Therefore, a more robust operation could be obtained when monitoring these parameters. The combination of Illumina together with operational data and stratification studies allowed the establishment of a link between the population structure and function of the anaerobic communities in the UASB reactor under certain conditions tested.

## Figures and Tables

**Figure 1 ijerph-18-01305-f001:**
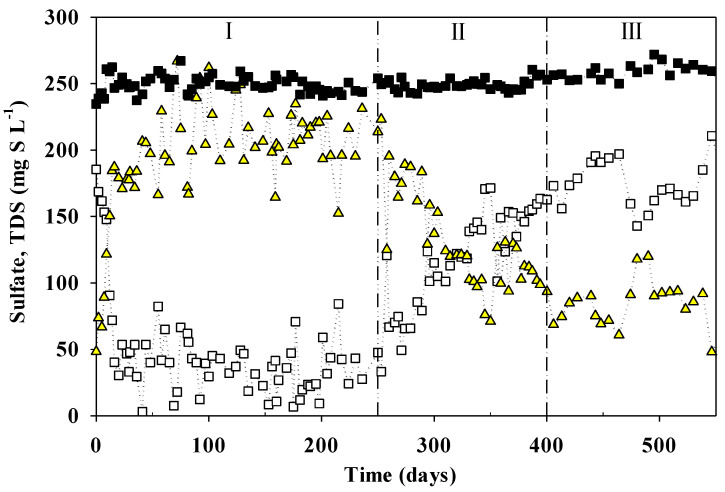
UASB performance during the long-term operation at constant sulfate loading rate. Sulfate concentration in the influent (■) and in the effluent (□), total dissolved sulfide concentration in the effluent (
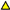
). Vertical lines represent the different periods.

**Figure 2 ijerph-18-01305-f002:**
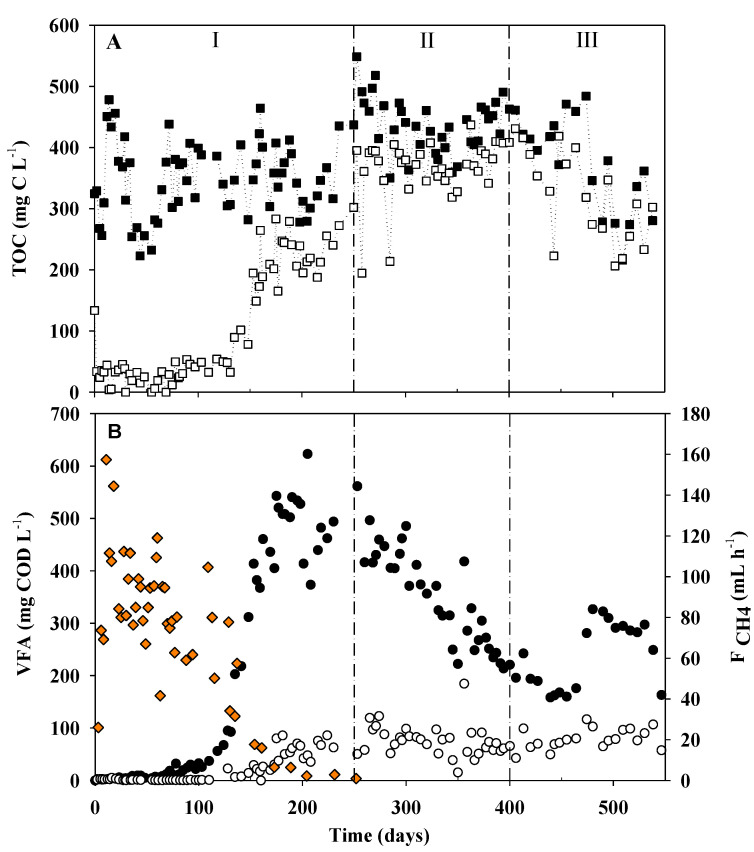
Performance of UASB. (**A**) TOC in the influent (■) and in the effluent (□). (**B**) Volatile fatty acids (VFA) concentration: acetic acid (●) and propionic acid (○) and flow of methane in the gas phase (
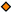
). Vertical lines represent the different periods considered.

**Figure 3 ijerph-18-01305-f003:**
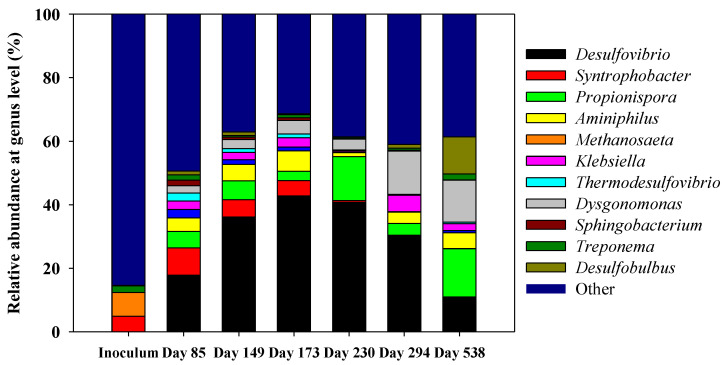
Relative abundances (%) at genus level of all the different samples along the operation of the reactor.

**Figure 4 ijerph-18-01305-f004:**
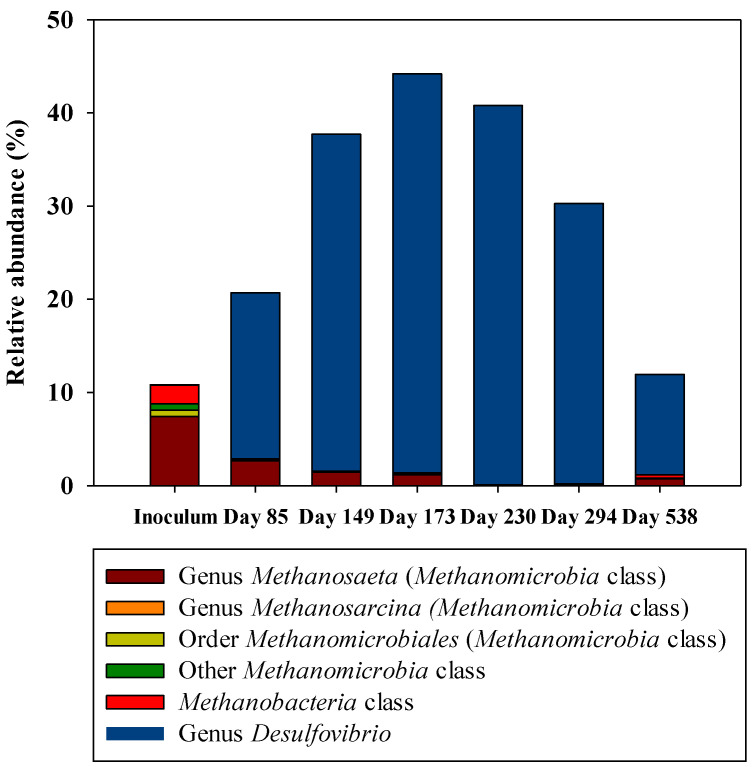
Relative abundances (%) of all the methanogens detected in all the samples together with the main sulfate reducer (genus *Desulfovibrio*).

**Figure 5 ijerph-18-01305-f005:**
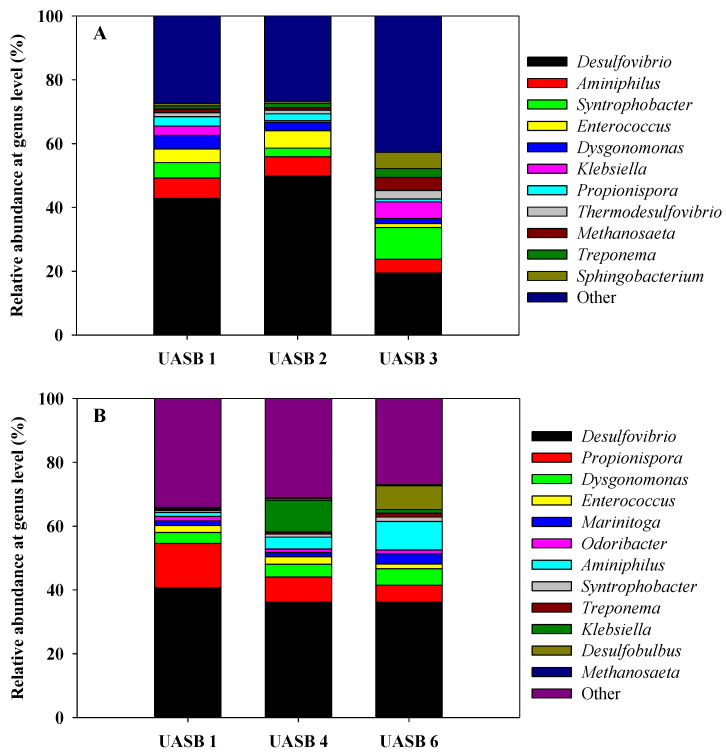
Microbial diversity along the sludge bed heights at the genus level. (**A**) Samples were collected on day 173 of the reactor operation. (**B**) Samples were collected on day 230 of the reactor operation.

**Figure 6 ijerph-18-01305-f006:**
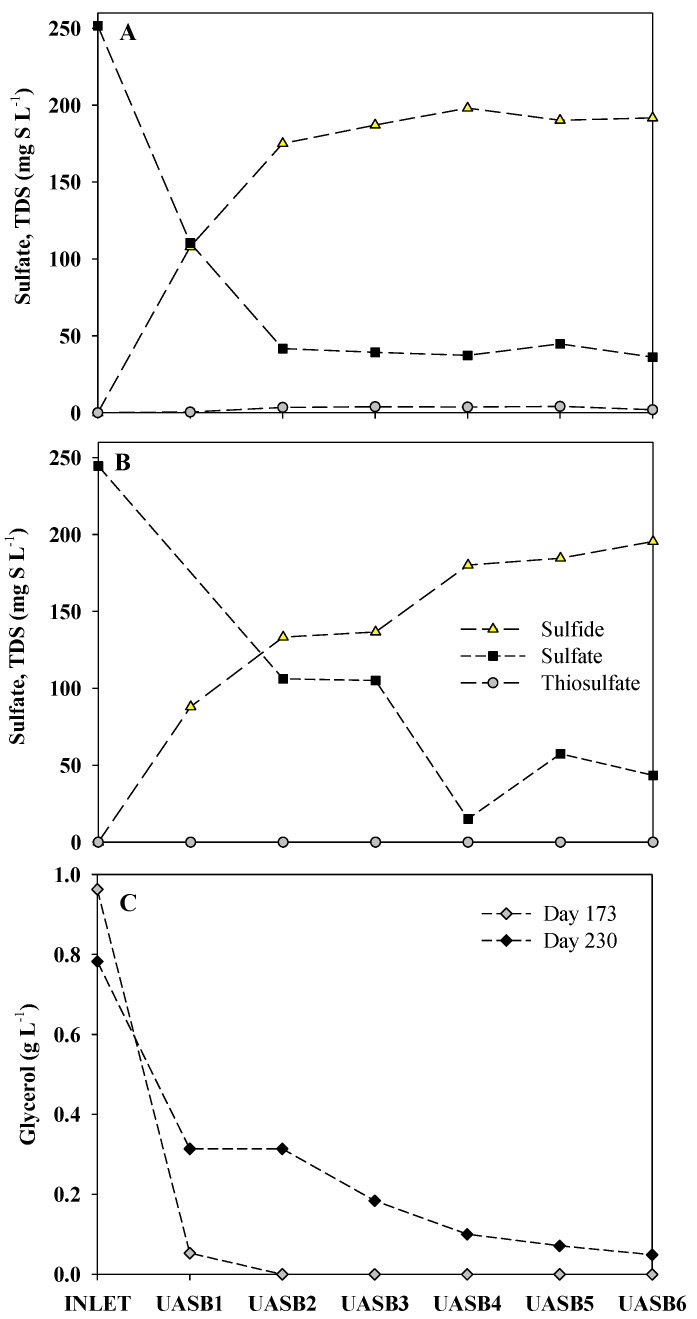
Experimental profiles obtained from the different reactor heights. (**A**) Sulfur species concentrations measured along the reactor´s heights on day 173. (**B**) Sulfur species concentrations measured along the reactor´s heights on day 230. (**C**) Glycerol concentration measured along the different reactor´s heights on days 173 and 230.

**Table 1 ijerph-18-01305-t001:** Biomass sampling events during the long-term operation of the UASB reactor.

Day of Operation	UASB Height
85	UASB 6
149	UASB 1,2,3
173	UASB 1,2,3
230	UASB 1,4,6
294	UASB 1,2,6
538	UASB 6

**Table 2 ijerph-18-01305-t002:** Removal efficiencies and elimination capacities were obtained along with the UASB operation.

Period	Days	TOC/S (g C g^−1^ S)	S-EC (kg S m^−3^ d^−1^)	TOC-EC (kg C m^−3^ d^−1^)	S-RE (%)	TOC-RE (%)
I	0–250	1.4 ± 0.3	4.4 ± 1.3	5.3 ± 2.1	80 ± 17	70 ± 27
II	250–400	1.7 ± 0.3	2.5 ± 0.7	1.4 ± 1.1	52 ± 15	16 ± 11
III	400–550	1.4 ± 0.3	1.6 ± 0.4	1.3 ± 1.1	33 ± 8	16 ± 11

**Table 3 ijerph-18-01305-t003:** Relative abundances (%) of methanogens and sulfate reducers (genus *Desulfovibrio*) on the different samples throughout the operation of the UASB reactor.

Sampling Time (Day)	Methanogens (%)	SRB (Genus *Desulfovibrio*) (%)
0 (Inoculum)	10.8	0.0
85	2.9	17.8
149	1.5	36.2
173	1.4	42.8
230	0.0	40.7
294	0.2	30.1
538	1.2	10.8

## Data Availability

Data is contained within the article.

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
