# Peer review of "Microbial Diversity Dynamics in a Methanogenic-Sulfidogenic UASB Reactor"

_ijerph, 2021, doi:10.3390/ijerph18031305_

Round 1

Reviewer 1 Report

The paper deals with the microbiology of a UASB reactor operating at constant conditions aimed at sulfate reduction in a SOx emissions treatment. The authors studied microbial communities during the operation time and related them to changes in operational results and chemical species.

In my opinion, the paper is very interesting and reveals the influence of microbial communities on the operation and the efficiency in carbon and sulfate removal.

The introduction is clear, concise, and shows all relevant information needed to understand the research topic. The Materials and Methods section also shows all the necessary information to understand the methodology and experimental setup used to perform the assay. Results are also very informative, easy to understand, and with detailed information.

The discussion section is also very clear and almost all important aspects of the experimental results are discussed. However, there is some information that may not be necessary for this paper (e.g. lines 443-444 regarding Syntrophobacter (S. wolinii)) since it does not give valuable information. On the other hand, in my opinion, some interesting aspects are not discussed sufficiently in the paper, and it may be interesting to include the author's insights on this. Some of the aspects that could be included in the discussion section to enrich the paper are the following:

  • The possible influence of operational conditions on the operational results: e.g. Could a lower load avoid the operational failure? Should be the ratio C/S changed to make the system work properly? Other recommendations for operating such a reactor to avoid failure.

Authors could go deeper into this discussion using also their previous research on this topic.

  • Different microbial profiles in reactor height could be also related to the mode of operation of a UASB reactor and can be also discussed.

If further analyses are planned regarding this experimental assay the slime attached to the reactor should be analyzed to get an appropriate insight on its origin, which could also help to explain reactor failure in a more precise way.

Reviewer 2 Report

The manuscript ijerph-1072356 is very interesting and well written. It is considered a valuable contribution for the field of environmental engineering, however there are some important issues that require clarification before being suitable for publication. For an objective assessment of this work, the authors should provide data about reactor pH, biogas H2S content, glycerol origin and composition, TSS and VSS concentration inside the sludge bed.

Specific comments

Keywords: Flue gas treatment seem not be clearly related with the manuscript content

Introduction:

The discussion provided by the authors about flue gas treatment and relevant applications seems not relevant to the manuscript content. We advise the authors to include a relevant paragraph in the discussion section describing possible applications for the proposed system/ biotechnology (i.e. for flue gas treatment, etc).

The introduction could benefit by a critical discussion of previous studies on the anaerobic treatment of sulfate-rich effluents and the potential toxicity of H2S on anaerobes.

Results

Data about biogas production and composition and especially the biogas H2S content are missing from this work. These data can be included in supplementary material.

The slime reported by the authors is not clear for the reader (considering the photograph provided in supplementary material). Moreover, it seem like sulfur deposits. Since the authors used tap water for the preparation of the synthetic medium, did they apply any chemicals to remove dissolve oxygen? Is it possible that DO could result in sulfide oxidation inside the reactor, considering also the relative short HRT applied (1-2 h)? Please clarify in the M&M section and in the overall discussion.

Figure 6: The concentration of glycerol at the effluent seem to be <100 mg/L (on day 230) however based on Figure 2a the concentration of TOC at the effluent seems to be around 250 mg/L. What is wrong here? Please clarify.

Data about reactor pH are missing from this work. This is important since pH can affect the balance between different sulfide species and enhance toxicity to anaerobes. These data can be included in supplementary material.

Line 419: the authors reported about impurities contained in crude glycerol. It is not clear if the authors used analytical grade glycerol or a by-product from the biodiesel industry? In the second case, glycerol may contain large quantities of lipids (HEM), LCFA methyl-esters and polyphenols that can cause toxicity to anaerobes. The authors are advised to provide an explanation about glycerol origin or data about composition and degradability throughout the experimental period.

TSS – VSS concentration inside the UASB reactor (and especially the sludge bed zone) are missing from this work. This is important to assess possible biomass washout related process deterioration. These data can be included also in supplementary material.

Did the authors noticed any sludge flotation and biomass losses during reactor operation?

The toxicity of H2S to methanogens and synthrophic bacteria is missing from the discussion and the introduction section.

Conclusions: The conclusions section does not reflect the content of this work. The authors should revise appropriately while also considering the comments provided above.

Round 2

Reviewer 2 Report

The manuscript is revised as per reviewer suggestions and is now suitable for publication